# Influence of Tool and Welding Parameters on the Risk of Wormhole Defect in Aluminum Magnesium Alloy Welded by Bobbin Tool FSW

Milan Pecanac [1], Danka Labus Zlatanovic [1,2], Nenad Kulundzic [1], Miroslav Dramicanin [1], Zorana Lanc [1], Miodrag Hadzistević [1], Slobodan Radisic [3,*] and Sebastian Balos [1]

[1] Department of Production Engineering, Faculty of Technical Sciences, University of Novi Sad, 21000 Novi Sad, Serbia; pecanac.milan@uns.ac.rs (M.P.); danlabus@uns.ac.rs (D.L.Z.); kulundzic@uns.ac.rs (N.K.); dramicanin@uns.ac.rs (M.D.); zoranalanc@uns.ac.rs (Z.L.); miodrags@uns.ac.rs (M.H.); sebab@uns.ac.rs (S.B.)
[2] Department of Production Technology, Technische Universität Ilmenau, 98693 Ilmenau, Germany
[3] Faculty of Economics and Engineering Management, University Business Academy Novi Sad, 21107 Novi Sad, Serbia
* Correspondence: sradisic@uns.ac.rs; Tel.: +381-21-4852435

**Abstract:** Bobbin tool friction stir welding (BTFSW) utilizes a special tool that possesses two shoulders interconnected by a pin instead of one: the top shoulder and the pin in the conventional FSW tool. This greatly simplifies the kinematics in the otherwise complicated setup of FSW since the bottom shoulder forms the bottom surface of the weld, without the need for a backing plate. Moreover, the tool enters the base metal sideways and travels, forming the joint in a straight line while rotating, without the need for downward and upward motion at the beginning and end of the process. This paper presents a study on the BTFSW tool geometry and parameters on the risk of wormhole defect formation in the AA5005 aluminum–magnesium alloy and the wormhole effect on mechanical properties. It was shown that higher stress imposed by the tool geometry on the joint has a significant influence on heating, an effect similar to the increased rotational speed. Optimal kinematic and geometrical tool properties are required to avoid wormhole defects. Although weld tensile strengths were lower (between ~111 and 115 MPa) compared with a base metal (137 MPa), the ductile fracture was obtained. Furthermore, all welds had a higher impact strength (between ~20.7 and 27.8 J) compared with the base material (~18.5 J); it was found that the wormhole defect only marginally influences the mechanical properties of welds.

**Keywords:** bobbin tool; friction stir welding; mechanical properties; wormhole defect

## 1. Introduction

Friction stir welding (FSW) is one of the most recently developed welding processes, patented in 1991 at The Welding Institute (TWI) in the U.K. [1–3]. It is a solid-state process that does not involve the melting of the material, and there is no need for additional consumables (fluxes and filler metals) for weld formation, nor does it release any excessive fumes or gases [1,2]. Not needing filler metals has one more advantage, namely, raw aluminum is not only complicated to manufacture, but it is evermore becoming increasingly scarce [4–7]. Moreover, environmental impact is significant, because $CO_2$ emission during aluminum ore processing is very high [8,9]. This fact points to many advantages of FSW from an environmental point of view. FSW uses a non-consumable tool for the formation of the weld material that is softened and mixed as a result of the action of the tool [10]. The heat necessary for the process is generated by the tool rotation and clamping (usually downward force) of the tool on the joint line. Material is softened and mixed as a result of the joint action of the tool rotation and transition alongside the joint line [10–12]. The quality of the welded surface and weld itself is dictated by the parameters applied and

tool geometry. The tool has two functions: (1) heating the material and (2) containing and directing the softened/plasticized base material of the weld [10]. Being a solid-state welding process, FSW enables the welding of dissimilar materials, e.g., different alloys with the same base metal, such as AA5052 to AA6061 [11], and dissimilar materials such as aluminum with copper, magnesium, or steel [13–17]. FSW, since it was patented, has shown great potential in further development of process variants, from the point of view of welding of different materials, base material thicknesses, and other specific aims, such as technological advantages. In Figure 1, a detailed classification of the friction stir welding process with emphasis on self-reacting FSW for various applications is listed.

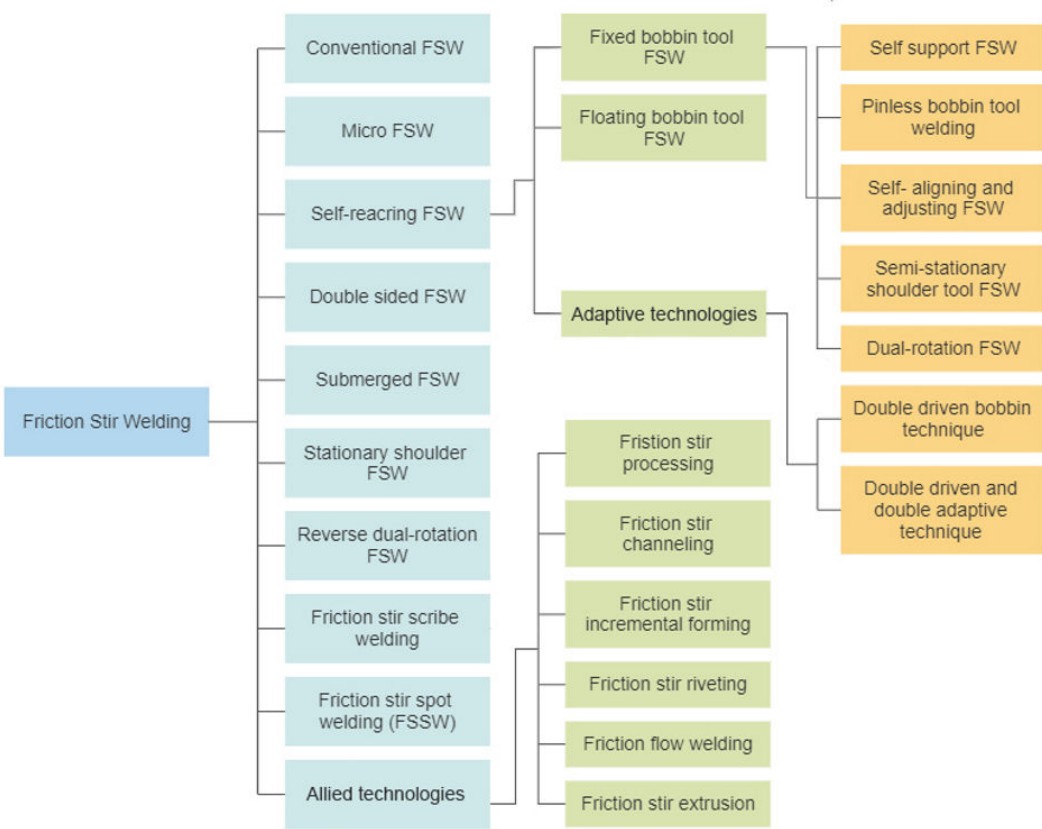

**Figure 1.** Schematic classification of friction stir welding processes, adapted with permission from [18]. 2020, Danka Labus Zlatanovic.

One special type of FSW process variant is the bobbin tool friction stir welding (BTFSW), also called self-reacting friction stir welding (SRFSW) [1,7,19–21]. BTFSW uses a special, unconventional tool design with a different tool geometry compared with the conventional FSW presented in Figure 2. The BTFSW tool has an upper and lower shoulder, enabling welding of workpieces throughout the whole thickness, thus eliminating the possibly unwelded bottom of the joint by clamping it between the shoulders. The unwelded bottom has an influence on the mechanical properties of the weld, especially on bending properties. Balos et al. [22] studied the influence of conventional friction stir welding of an aluminum–magnesium alloy (AA 5052) on bending properties. It was found that all samples except one with the lowest welding speed (17 mm/min) did not withstand bending up to 180° without crack. In all specimens, the angle of the bend to the first crack was lower than 70°. This alloy (AA 5052), as well as others from series 5xxx, show better mechanical properties when they are welded with solid-state welding techniques, such as friction stir welding, compared with the fusion welding techniques. Even though FSW provides joints without pores, hot cracks, and residual stresses, bending properties, especially over the root, are poor. This is why a new concept with a double shoulder (bobbin tool) was proposed to

avoid initial cracking due to an unwelded bottom side. This feature is also very suitable for the welding process because it does not require any downforce like conventional FSW. Moreover, the whole welding setup is simplified, because BTFSW is performed without the need for the bottom-supporting backing plate [23].

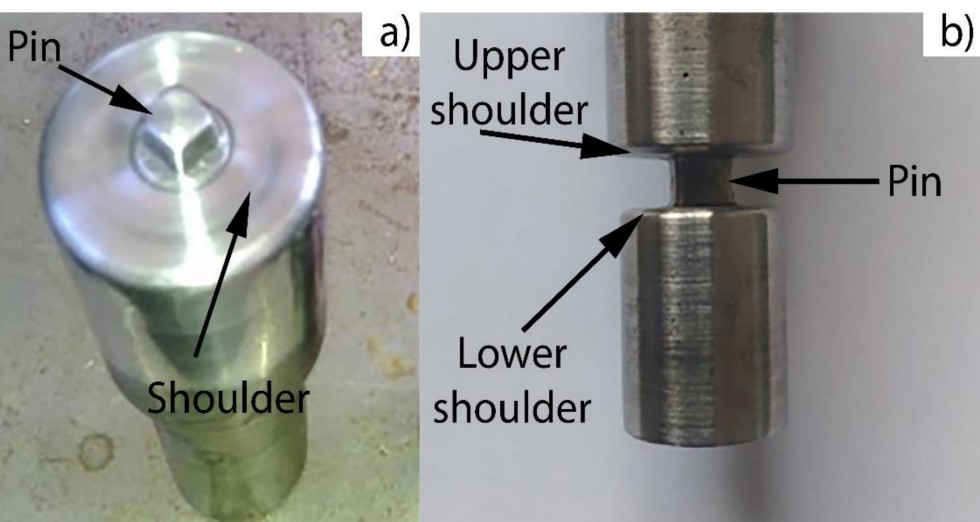

**Figure 2.** FSW tools: (**a**) conventional FSW tool; (**b**) BTFSW.

BTFSW has been the subject of fewer studies compared with the conventional FSW. In the work by Wen et al. [24], BTFSW was successfully applied mostly for the welding of aluminum (series 2xxx) base metal. It was shown that mechanical properties are defined by the welding parameters. They achieved joint efficiency of around 70% of the base material. Moreover, the hardness distribution throughout the thickness of the weld is symmetrical, and the lowest hardness was found in the heat-affected zone (HAZ). Wang et al. [25] researched the effect of tool rotation speed on the microstructure of the AA2198 alloy, concluding that with an increase in rotational speed from 400 to 800 rpm, joint tensile strength increased as well, and then decreased with the increase in the rotation speed from 800 to 1000 rpm. They also achieved joint efficiency of up to 80% of the base material. Chumaevskiia et al. [26] have used a bobbin tool with a round pin and thread for welding aluminum AA 2024 alloy. It seems that this tool suffered a high load which led to breaking of the tool, stressing one drawback of this process. Fuse et al. [27] also used a round bobbin tool with three flats and three different shoulder diameters (20 mm, 22 mm, and 24 mm) to weld the aluminum AA 6061-T6 alloy. The tool with a diameter of 24 mm provided the highest mechanical properties due to the highest temperature introduced in the base metal.

Asadi et al. [25] used different pin geometries to determine the impact on the weld joint. They used different welding speeds and tool rotation speed was constant, and it can be seen that a square-shaped pin produces one of the finer grain sizes and smaller traverse force through the welding process. They also concluded that when the pin has multiple edges (polygonal shape), the temperature increases. Ahmed et al. [28] also used multiple pin shapes for welding in order to develop a mathematical model for heat input. They concluded that the amount of heat generated during BT-FSW is directly proportional to the tool rotation rate, friction coefficient, welded pate thickness, shoulder radius, shoulder concave angle, and the tool pin surface area at constant shoulder surface areas. Meanwhile, it is inversely proportional to the tool travel speed. In all presented research, one correlation can be derived: the higher the induced temperature in the workpiece, the higher the weld mechanical properties are. The main aim of this study was to find the influence of two different shoulder angles of BTFSW tools having the square shape pin on welding temperatures, microstructural features, and mechanical properties. Moreover, parameter optimization was performed to find the optimal combination of tool geometry and process parameters. Temperatures were measured during the welding process and the highest

temperature was reported and correlated to mechanical properties and imperfection occurrence. The maximal temperature was obtained and its influence on the tools during the process as well as mechanical properties of the welds such as tensile and impact strength and bending properties were evaluated. In addition, macro and microstructural studies were performed to find the influence of the tool shoulder angle.

## 2. Materials and Methods

Experimental work was performed on the AA5005 H32 alloy plates with specimen dimensions 140 mm × 60 mm × 4.8 mm and the chemical composition presented in Table 1. The chemical composition of the base material was obtained by ARL 3580 (Thermo Scientific, Waltham, MA, USA) optical emission spectrometer (OES). Tensile properties were obtained by ZDM 5/91 (VEB, Leipzig, Germany) tensile testing machine and they are provided in Table 2.

**Table 1.** Chemical composition of the base material (mass. %).

| % | Cu | Mn | Mg | Si | Fe | Zn | Ti | Al |
|---|---|---|---|---|---|---|---|---|
| Base material | 0.07 | 0.13 | 0.59 | 0.23 | 0.28 | 0.055 | 0.02 | Balance |

**Table 2.** Mechanical properties of the base material.

| Rp (MPa) | Rm (MPa) | A (%) | Z (%) |
|---|---|---|---|
| 119 | 137 | 16 | 61 |

Welding process was performed on the vertical milling machine FSSGVK-3 (Prvomajska, Zagreb, Croatia), adapted for the friction stir welding process, by the addition of a specially designed clamping system shown in Figure 3. The clamping system was designed to firmly clamp base metal sheets and allow the movement of the lower shoulder along the contact surface. In order to perform the welding process, two self-reacting tools were designed as shown in Figure 4. Tools were made of AISI H13 (EN X40CrMoV5-1) hot work tool steel, heat-treated by oil quenching and tempering to obtain a hardness of 53 HRC. The main difference between the two tool types was the shoulder angle which was set at 2° and 4°. The main goal of using the two shoulder angles is to determine which of the two has greater influence on the weld, from of plastic deformation point of view. A square-shaped pin was used because previously mentioned literature stated that this shape has a smaller transverse force during the welding and generates greater heat input. The maximal temperature was obtained and its influence on the tools during the process as well as mechanical properties of the welds, such as tensile and impact strength and bending properties, were evaluated. In addition, macro and microstructural studies were performed to find the influence of the tool shoulder angle.

The weld designation system is presented in Table 3 and the specimen cutting plan is shown in Figure 5.

**Table 3.** Tools and parameters designation system.

| Weld Designation | Tool Type | Tool Shoulder Angle (°) | Tool Rotation Speed (rpm) | Welding Speed (mm/min) |
|---|---|---|---|---|
| A2-1 | | | 900 | |
| A2-2 | A2 | 2 | 1120 | |
| A2-3 | | | 1400 | |
| A4-1 | | | 900 | 20 |
| A4-2 | A4 | 4 | 1120 | |
| A4-3 | | | 1400 | |

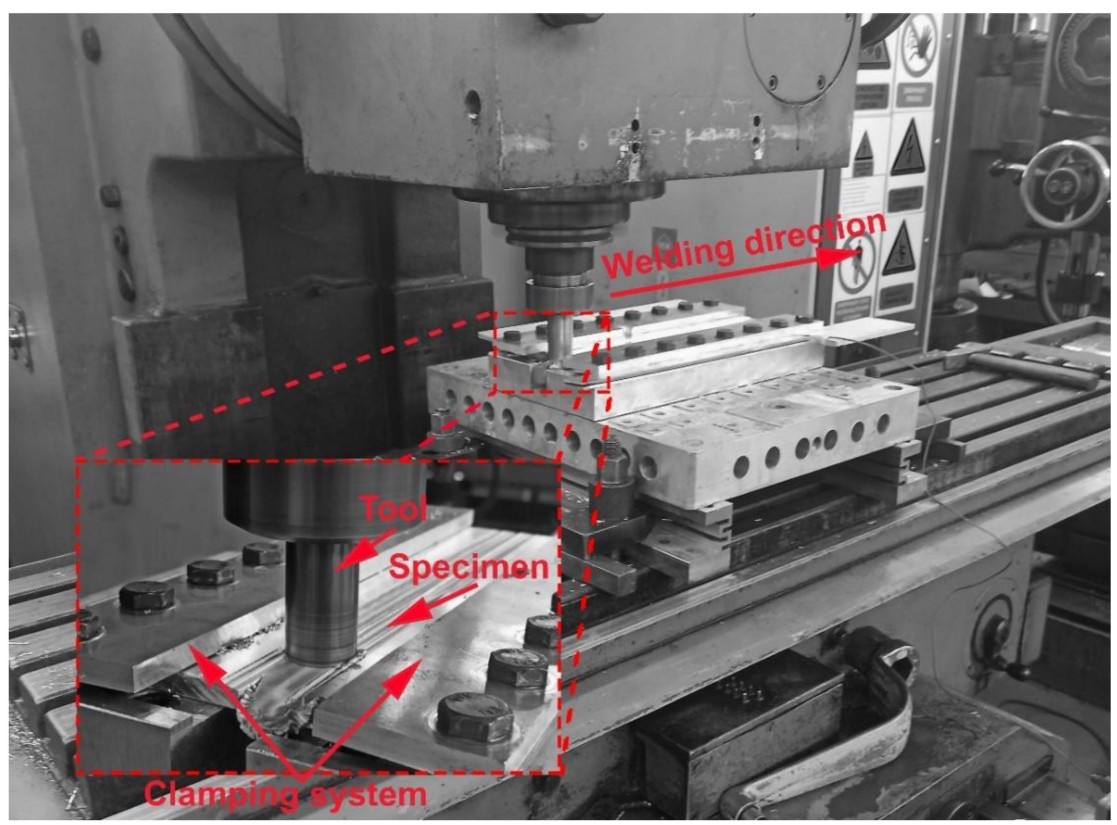

**Figure 3.** Bobbin friction stir welding process set-up.

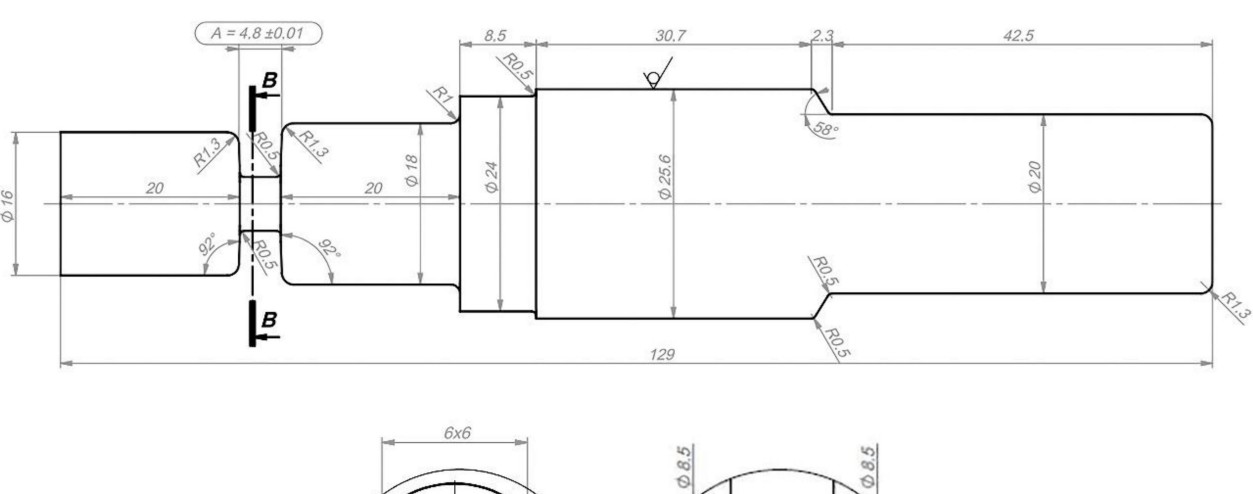

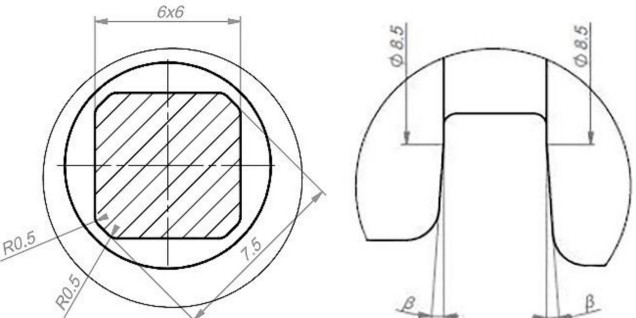

**Figure 4.** Tools used: in tool A2, $\beta_1 = 2°$; in tool A4, $\beta_2 = 4°$ (All units in are in milimeters).

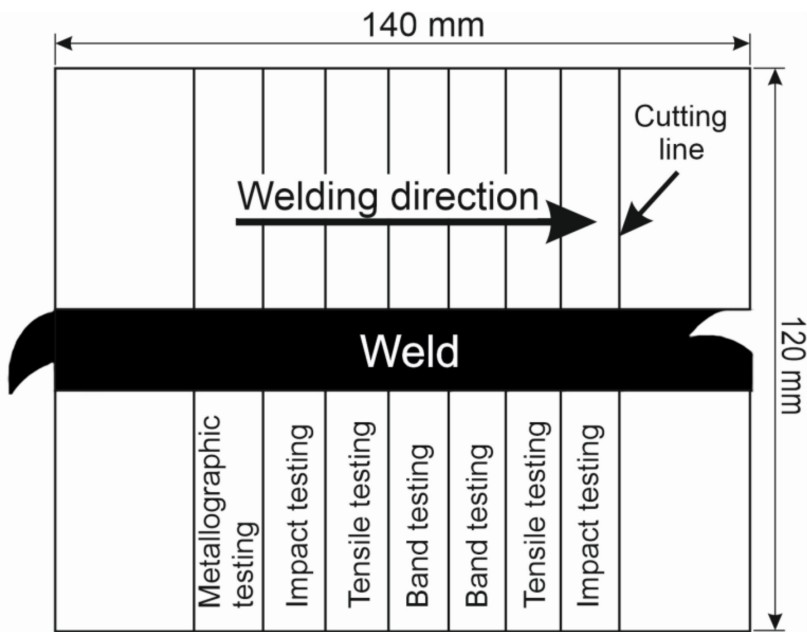

**Figure 5.** Specimen cutting plan.

Testing of the welded joint included: tensile testing, three-point bend test, Charpy impact test, and macro analysis. Tensile testing was carried out in accordance with EN ISO 4136:2012 standard on a VEB ZDM 5/91 testing machine. Ultimate tensile strength, proof strength, and cross-section reduction were measured, and average values were reported. The Charpy impact test was performed in accordance with EN ISO 148-1:2016 standard on an instrumented Charpy pendulum JWT-450 (Jinan, China), at room temperature. A standard V-notch in the Charpy specimens was placed in the nugget zone (NZ). A three-point bend test was also performed on the VEB ZDM 5/91 machine in accordance with EN ISO 5173:2009 in two specimens: one over the top and the other over the bottom of the weld. Bending was carried out until a crack appeared on the surface of the sample and the angle was reported. Afterward, the testing was continued until fracture. If the crack or fracture did not appear, the bending was conducted until reaching 180°.

Metallographic samples were prepared by grinding (abrasive papers from 150 to 2500 grit) and polishing (suspensions with 6, 3, 1, and $\frac{1}{4}$ μm diamond particles), and finally, etching was done with a solvent of 1 mL of hydrofluoric acid and 24 mL of distilled water. Weld defects were analyzed with a light microscope. The light microscope used was Leitz Orthoplan.

## 3. Results and Discussion

### 3.1. Metallographic Examinations

Macrographs of welded samples are presented in Figure 6. A distinct hourglass shape of the welded sample typical for the BTFSW joining process can be observed, containing the common nugget zone (NZ), heat affected zone (HAZ), and thermomechanically affected zone (TMAZ). The microstructure of the base metal (BM) is shown in Figure 7a, with distinct uniaxial grains. Highly refined predominantly uniaxial grains are present in the NZ, Figure 7b. The transition TMAZ/HAZ zone with elongated grains is shown in Figure 7c,d.

Visual examination in Figure 5 reveals the existence of wormhole defects (tunneling) in specimens A2-1, A2-3, and A4-2. In specimen A2-1, in the wormhole there are micro defects (0.018 mm² and 0.011 mm²), while in A2-3 (0.311 mm²) and A4-2 (0.269 mm²), the wormhole defect is a relatively large, triangular in shape. In specimen A4-2, the wormhole extends almost to the bottom of the specimen in the form of a crack, indicating an unbalanced kinematic factor of the tool rotation speed and welding speed. In specimens A2-1, A2-3,

and A4-2, the wormhole is on the advancing side of the weld, which is in accordance with the results obtained in [22,29,30]. In specimens A2-2, A2-3, and A4-3, an intensive flash was observed on both the top and bottom surfaces.

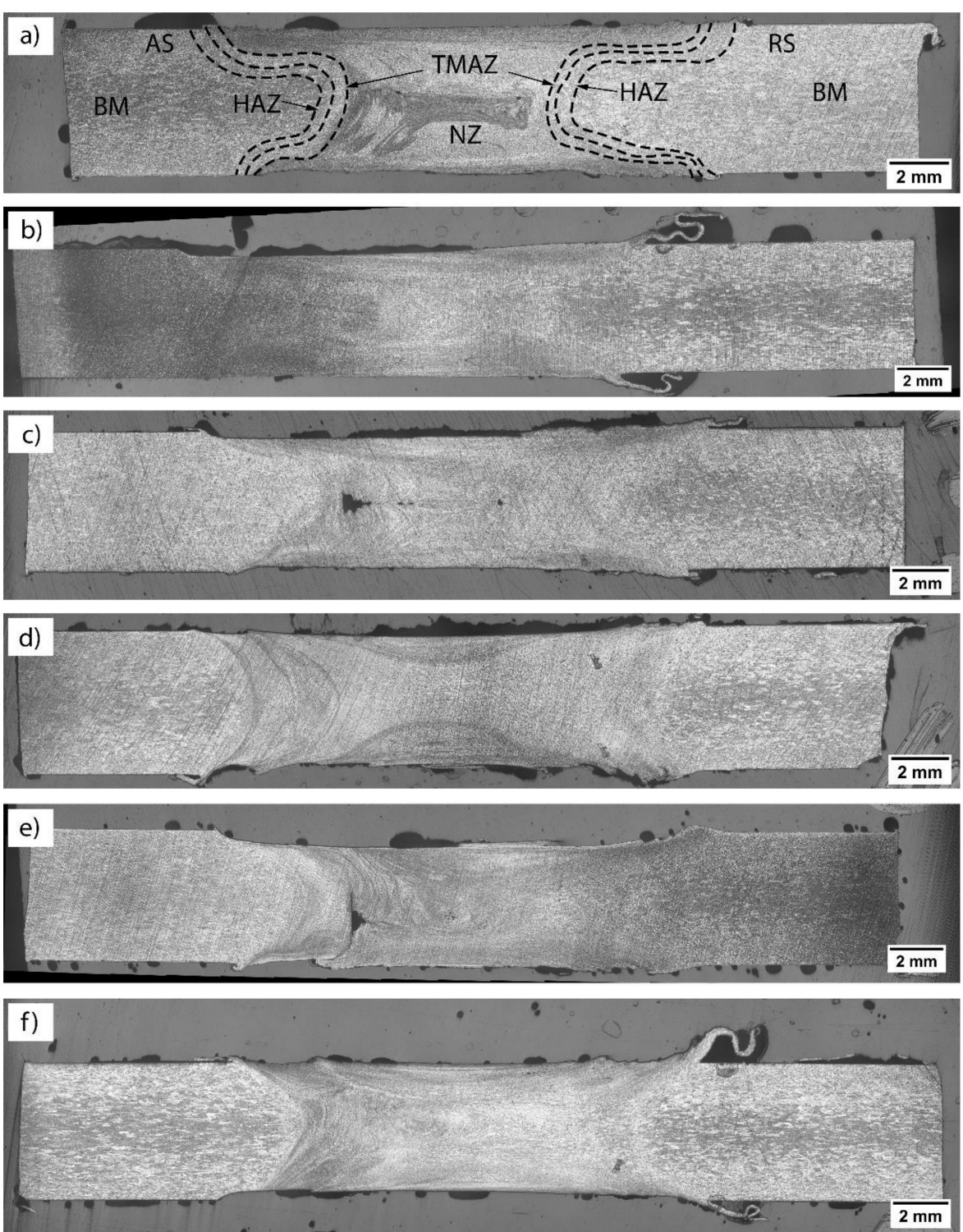

**Figure 6.** Cross-sectional macrographs of specimens: (**a**) A2-1; (**b**) A2-2; (**c**) A2-3; (**d**) A4-1; (**e**) A4-2; and (**f**) A4-3.

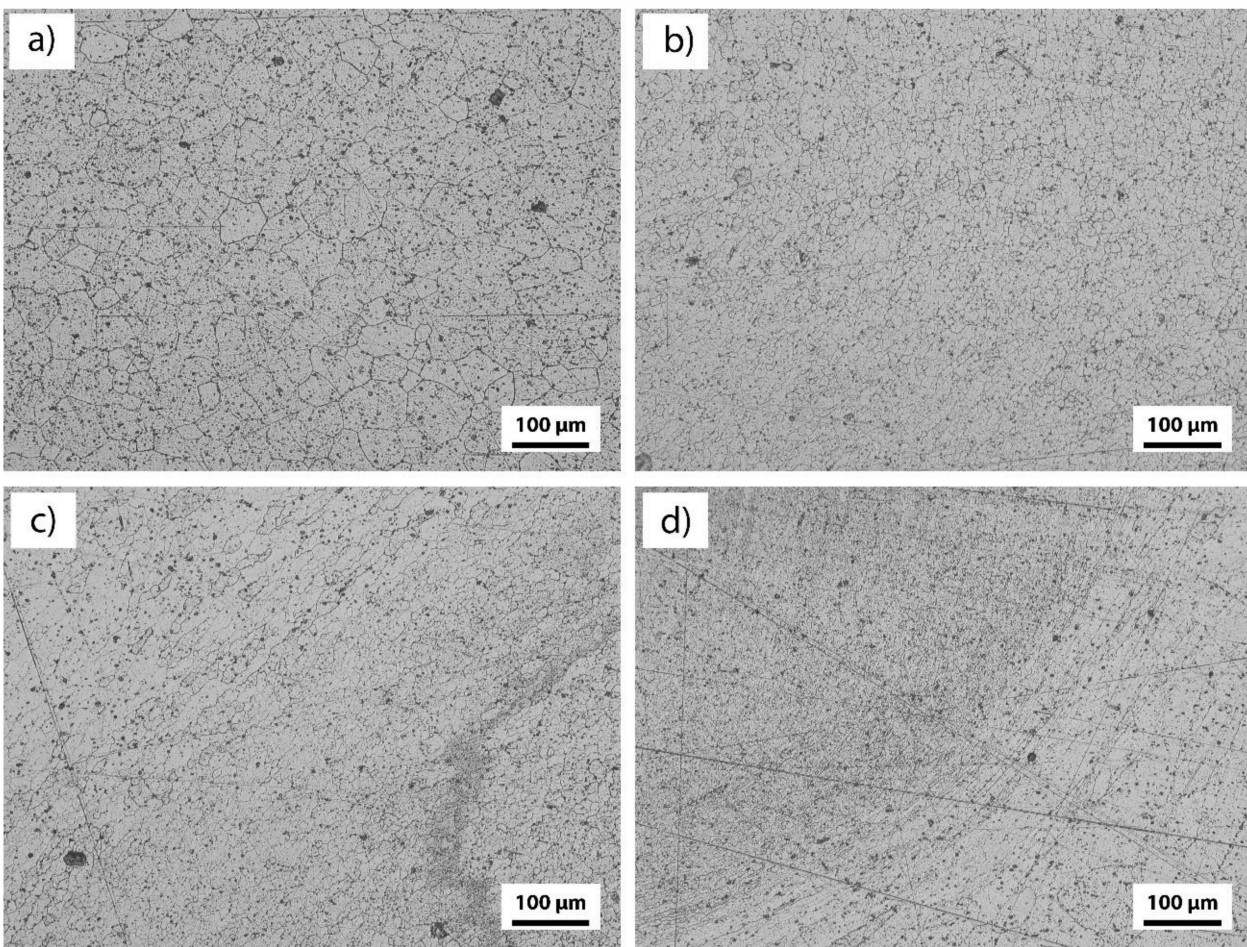

**Figure 7.** Microstructure of the base material (**a**), nugget zone (**b**), and two transition zones (**c**,**d**) in specimen A4-3.

*3.2. IR Thermography*

Thermograms showing the maximum temperatures detected in the welding process are shown in Figure 8. Maximum temperature in the process was detected in the area comprising the tool shoulder and base material surface. The temperature increases as the tool rotation speed is increased, due to increased friction. Moreover, tool A4 with shoulders angled at 4° possesses a higher heat-generating potential compared with the A2 tool with shoulders angled at 2°. This is due to a higher pressure imparted by the tool shoulders with a larger angle of the shoulder, which induces higher internal friction and heat. Moreover, the difference between maximum achieved temperatures becomes lower as the tool rotation speed is higher, indicating that the tool rotation speed has the primary influence on the overall heat generation during BTFSW. From Figure 8, it can be seen that temperature intervals of the tools A2 and A4 are in the given range from 575.9 to 606.9 °C and from 590.2 to 609.2 °C, respectively.

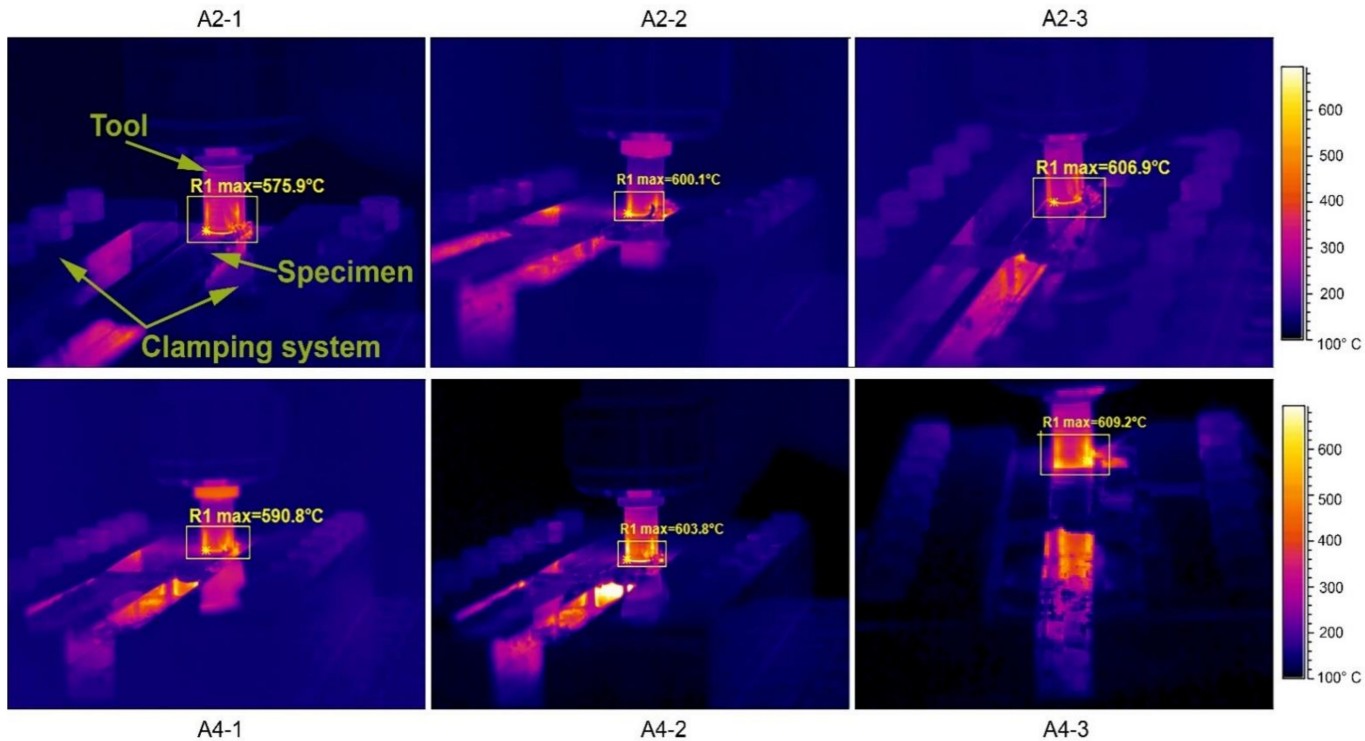

**Figure 8.** Thermograms of the BTFSW welding process.

### 3.3. Tensile Testing Results

Tensile testing results, standard deviations, and fracture location in terms of the FSW zone and the side are presented in Figure 9. The first listed value refers to the tensile specimen closer to the welding start, that is, the tool entrance in the weld zone. In the majority of specimens, the strength of the specimen closer to the entrance of the tool is marginally higher compared with the specimen taken from the section closer to the tool exit. This can be attributed to the effect of heat generated by the tool that is transmitted through the specimen and is more intense in the weld closer to the exit than at the beginning of the welding process, stressing the heat softening effect. The application of tool A4 results in higher tensile strengths compared with the specimens obtained with tool A2 due to a more intense deformation offered by the tool with a more conical geometry. The highest tensile strength was obtained with tool A4 and the rotating speed of 1120 rpm, closely followed by the specimen obtained with the same rotating speed and tool A2. All values are lower compared with the tensile strength of the base metal. Selected stress–strain charts are shown in Figure 9, revealing the ductile nature of the material, which is in accordance with the reductions of the area also listed in Figure 9. Fracture location is the zone between HAZ and TMAZ on the retrieving side (RS), except for specimens A2-3 and A4-2, which is the most critical part of the weld due to the lowered stirring effect (angular velocity of the pin and welding speed are subtracted). In specimens A2-3 and A4-2, the fracture occurred near the SZ/TMAZ at the advancing side (AS) where a significant wormhole is observed, having a size of around 1 mm. Despite this, tensile strength in specimen A4-2 is the highest of all tested. On the other hand, tensile strength in specimen A2-3 is the lowest, which indicates the possibility of an irregular wormhole in specimen A4-2, existing only in certain areas along the length of the weld. Finally, specimen A4-3, which was obtained without the tunnel (Figure 5), possesses a lower tensile strength compared with A4-2, due to a higher rotational speed of the tool which generates a higher amount of heat, as indicated in Figure 8. This is in contrast to specimens A2-2 and A4-1, where clearly, the heat input (Figure 8) has secondary importance compared with the work hardening effect. However, this is in accordance with the results of reduction in the area of these specimens,

as well as the results presented in [29]. Proof strengths follow tensile strength trends, while reductions of area are inversely proportional to strengths.

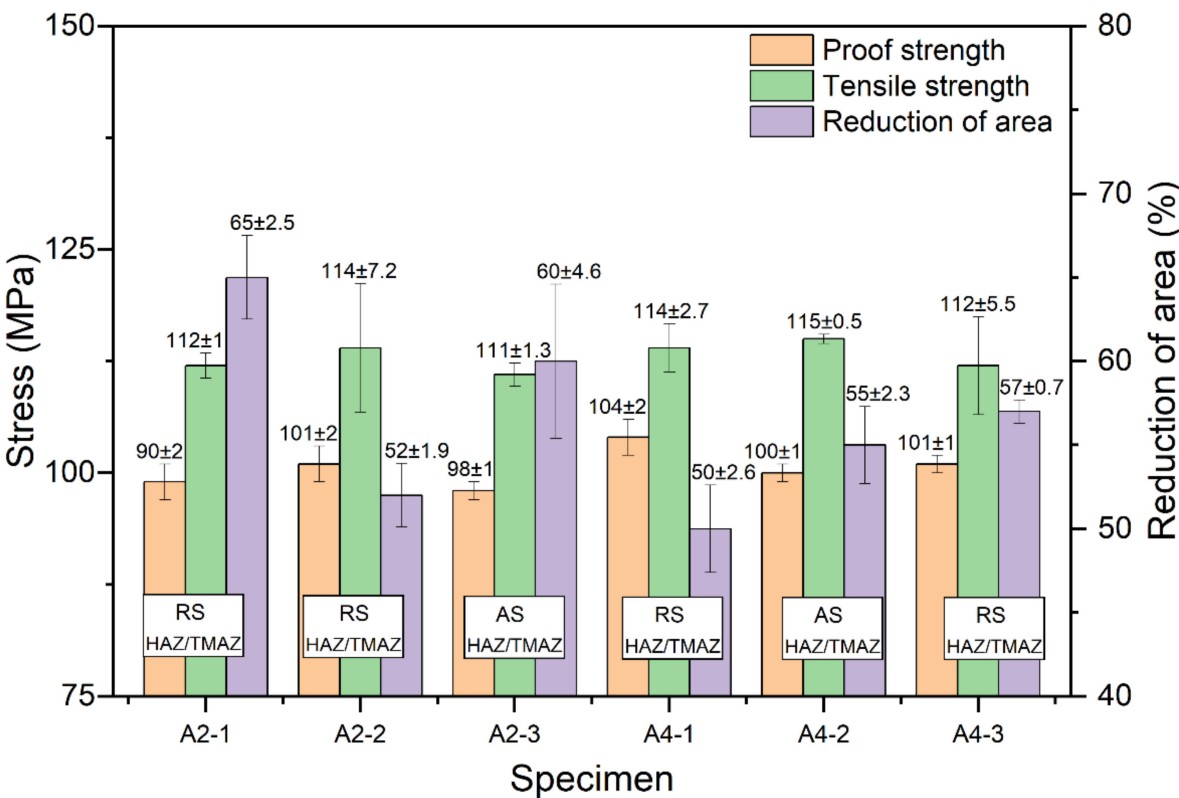

**Figure 9.** Results of tensile testing (AS—advancing side; RS—retrieving side; SZ—stir zone; HAZ—heat-affected zone; and TMAZ—thermomechanically affected zone).

Generally, by increasing the shoulder angle from 2° to 4°, proof and tensile strengths slightly increase, which is the result of predominant hardening over thermal softening due to higher heat input. Furthermore, reductions in area are inversely proportional to strengths. No clear indication of tool rotation speed influence can be derived, with a probable disturbing factor being the appearance of wormholes. A common tendency is that a higher tool rotation speed results in a predominant thermal softening over strain hardening [31].

Weld strength efficiencies are between 81 and 84% for all specimens, which is similar to other research results related to strain-hardened aluminum alloys [32]. However, in Al–Mg alloys with −0 temper condition, a higher weld strength can be obtained compared with base metals [22].

*3.4. Bend Test*

The results of the three-point bend test are presented in Table 4. Bending was carried out over the bottom and top surface of the weld, that is, surfaces formed by the top and bottom shoulder of the tool (Ø18 and Ø16 mm), respectively. It is evident that only specimen A4-2 bent over the bottom surface results in the appearance of a crack. This is the result of a wormhole shown in Figure 5e, extending almost to the bottom surface of the specimen. However, even though this specimen could have been bent to the maximum of 180°, it does not result in a fracture of the specimen. In other specimens, no cracks occurred during the execution of the bend test. Bend testing specimens are shown in Figure 10. These results indicate a clear advantage of the bobbin tool, which influences a simultaneous formation of both weld surfaces by the tool shoulders. This is in contrast to conventional FSW, where the weld face is formed by the tool shoulder, while the root is

formed by the backing plate. The weld root formation without defects is closely related to the distance between tool tip and backing plate and the maintenance of this distance is of utmost importance. Compared with our previous results using conventional FSW tools with complex concentric shoulders [22], there is a higher consistency in obtaining convenient bending properties without cracks.

**Table 4.** Bend test results.

| Specimen | Bending Over the Bottom Surface | | Bending Over the Top Surface | |
|---|---|---|---|---|
| | First Crack (°) | Bend Test to 180° | First Crack (°) | Bend Test to 180° |
| A2-1 | None | No fracture | None | No fracture |
| A2-2 | None | No fracture | None | No fracture |
| A2-3 | None | No fracture | None | No fracture |
| A4-1 | None | No fracture | None | No fracture |
| A4-2 | 14.3 | No fracture | None | No fracture |
| A4-3 | None | No fracture | None | No fracture |

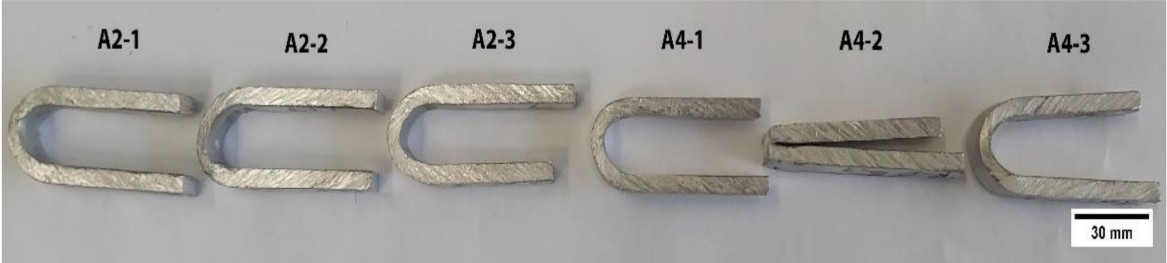

**Figure 10.** Bend testing specimens over the bottom of the weld.

*3.5. Charpy Impact Test*

The results of the Charpy impact test are presented in Figure 11. Crack initiation, crack propagation, and the impact energy (sum of the two) of the specimens and base metal are shown. It can be seen that the impact energy of all welds is higher than that of the base metal. It is evident that the existence of tunnels has an adverse influence on reaching high values of impact energies. However, the existence of micro tunnels (A2-1) does not have the same effect as the existence of larger tunnels in specimens A2-3 and A4-2. Nevertheless, specimens obtained with the tool having 2° shoulders (A2) have lower Charpy impact energy compared with specimens obtained with the tool with 4° shoulders (A4). Although A2-1 has relatively high impact energy, it is not as pronounced as the reduction in area during tensile test. It seems that the existence of a micro tunnel has a higher influence on the impact test than in the quasistatic test as a tensile test. The only exception is A2-2, with slightly higher impact energy; however, this can be attributed to the existence of a wormhole in specimen A4-2. The tool of a steeper shoulder (4°) imparts a higher pressure on the base material, increasing both the strain hardening component and heat, as shown by thermograms. Although higher rotation speed increases heat input, as shown by thermograms, it also decreases impact energies. In Figure 12, force-time charts of representative specimens are shown. The most significant difference between force to displacement charts obtained for the base metal and the A4-1 specimen is the lower maximum force in A4-1 and a different shape of the chart after the maximum force. In base metal, the shape is concave up, while in A4-1 it is concave down, both decreasing. This influences a higher crack propagation energy, which, along with a delayed maximum force, results in higher overall impact energy in the A4-1 specimen compared with the base metal. A similar tendency can be observed in other specimens.

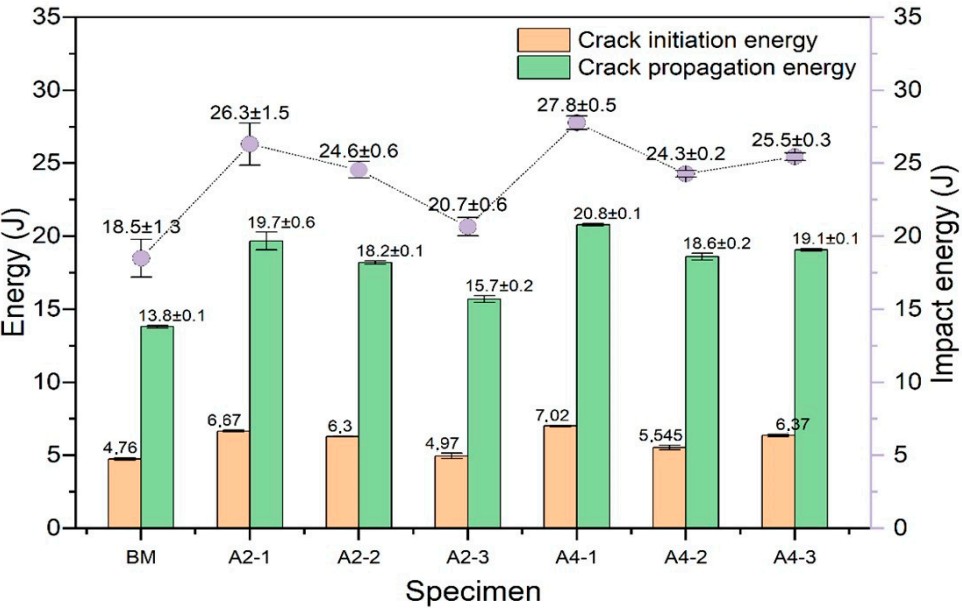

**Figure 11.** Charpy impact test results.

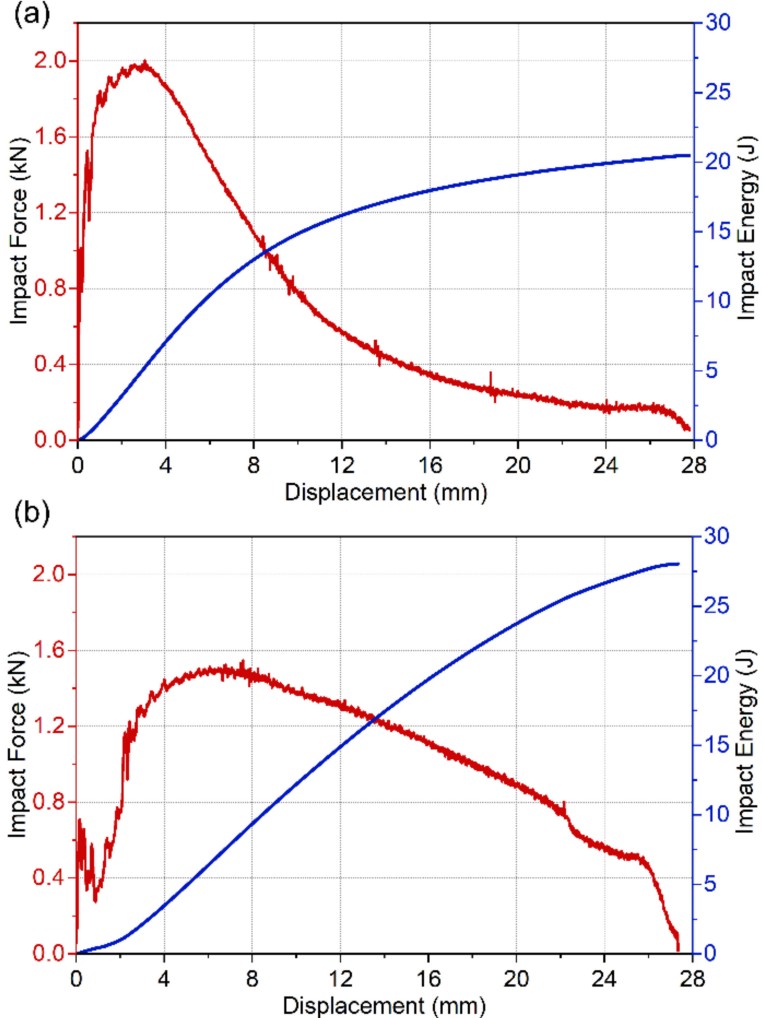

**Figure 12.** Force-time charts for (**a**) base metal; (**b**) specimen A4-1.

Overall, a steeper shoulder angle (4° versus 2°) is more aggressive in increasing heat generation potential, a decrease in wormhole risk occurrence, and resulting in generally higher mechanical properties. However, a larger shoulder angle also increases stresses in the tool itself, so, although this study did not take this into consideration, it can be assumed that a more stressed tool might have a higher risk of fracture and a shorter life.

## 4. Conclusions

In this study, Al–Mg alloy specimens were welded by using BTFSW with different parameters and two tool types, having 2° and 4° shoulder angles.

A more pronounced conical shape of shoulders induces higher temperatures during BTFSW, increasing the maximum temperature achieved. However, although higher heat input influences a higher thermal softening, in this case, a more predominant strain hardening occurred. This was shown by generally higher strengths and lower reductions in area in tensile testing specimens.

Wormhole defect occurs both with 2° and 4° shoulder angle, but at 4° it occurs at a lower tool rotational speed. Furthermore, there are indications that the wormhole is not continuous, indicating a larger shoulder angle is more beneficial as it increases the strain on the material. However, where it exists, the wormhole protrudes nearly to the bottom of the specimen, forming an incomplete joining that opens when the specimen is bend tested. The risk of failure is thus greatly increased. Moreover, higher stresses in the material also mean higher stress in the tool in terms of wear and fatigue.

Although tensile properties are lower (between ~111 and 115 MPa) compared with a base metal (137 MPa), the ductile fracture was obtained in all specimens.

All obtained impact strengths (between ~20.7 and 27.8 J) were higher compared with a base metal (~18.5 J), both in crack initiation and crack propagation energies, regardless of the existence of a wormhole defect. This is the result of grain refinement in the nugget zone.

The wormhole defect that does not extend into the incomplete joining only marginally influences the mechanical properties of welds.

**Author Contributions:** M.P., D.L.Z., M.D., N.K. and Z.L. performed the experiments; M.P. and S.B. wrote the paper; S.B. designed the experiments; M.P., S.B., D.L.Z. and M.H. provided the resources; S.R. reviewed and edited the manuscript and provided work administration and supervision. All authors have read and agreed to the published version of the manuscript.

**Funding:** This research received no external funding.

**Institutional Review Board Statement:** Not applicable.

**Informed Consent Statement:** Not applicable.

**Data Availability Statement:** Not applicable.

**Acknowledgments:** The authors gratefully acknowledge research support by the project entitled "Materials, joining and allied technologies" in the Department of Production Engineering, Faculty of Technical Sciences Novi Sad, Serbia.

**Conflicts of Interest:** The authors declare no conflict of interest.

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
