# Peer review of "Influence of Tool and Welding Parameters on the Risk of Wormhole Defect in Aluminum Magnesium Alloy Welded by Bobbin Tool FSW"

_metals, doi:10.3390/met12060969_

Round 1

Author Response

General remarks:

Response to reviewers

Review 1

First of all, we would like to thank you for finding the time to review our manuscript. We appreciate all comments and suggestions, and we hope we were able to address them properly.

General Comments: The article presents the results of a process and tool study for FSW using a fixed-shoulder bobbin tool. The study focusses on examining the results when the tool shoulder angles are changed from 2 degrees to 4 degrees. The study also attempts to optimize welding parameters.

There are many studies available regarding the optimization of welding parameters. Although welding parameters and tool geometry are interconnected, I would encourage the authors to discuss in greater detail their work on the study of tool geometry and its influence on weld quality. This would most likely be of greater interest to the welding community.

Thank you for your suggestion. We have adapted the discussion according to your comment.

Specific Comments to be addressed:

  • Figure 1 on page 2 is difficult to read. Please enlarge and discuss more when providing an overview of FSW. It is presented but the discussion if very limited. A discussion would give the reader a better perspective on how this work fits into the array of solid state welding processes and technologies.

Thank you for your remark. To make it clearer we have reduced the classification only on FSW. We realised that the solid-state welding – classification was maybe too extensive for this paper.

  • Page 3 lines 72-88. Additional references and discussion of prior studies regarding BTFSW are required. Although references [23 & 24] are good, more are needed for comparison to the presented work. Discuss how these are different from the present work. State with detail what value (or innovation) does the presented work provide.

Thank you for your suggestion. We have adapted the introduction according to your comment.

  • Page 4. Include a picture / figure of the milling machine used for this work. Label key comments of the setup in the picture / figure.
  •  

Thank you for your suggestion. Figure is added and labelled.

  • Discuss the advantages of the 2 degree and 4 degree should setup. Why is this desirable for a bobbin tool? Explain to the readers.

We added a paragraph at the end of the tensile testing section. Also, additional comments in Charpy test section.

  • Figure 7 on page 8. Label the tool and workpiece in the figure. Both are difficult to see.

Thank you. We have labelled the figure.

  • Page 11, Conclusions section. Please do not use bullet format. Discuss the results with detail and written in paragraph form.

Thank you. The correction was made.

  • Page 11, Conclusions, lines 261 and 262. Elaborate and discuss further the conclusions regarding the conical shape of the shoulder and the increase in mechanical properties. Seems like this should be a key finding of the study and warrants more discussion.

We added an explanation at the end of the Results and Discussion and in Conclusions.

  • Page 11, Conclusions, lines 261 – 265. The statements of welding temperature / heat contradict each other. In lines 261 – 261 it is stated that higher temperature leads to increasing mechanical properties. In lines 263 -265, it is stated that more heat from higher rotational speed results in lower mechanical properties. Please discuss further and explain.

We elaborated on this, thank you. We added the explanation that a higher heat generation influenced a lower risk of wormhole occurrence.

Reviewer 2 Report

Dear Authors,

I have reviewed your paper: "Influence of Tool and Welding Parameters on the Risk of Wormhole Defect in Aluminium Magnesium Alloy Welded by Bobbin Tool FSW".

It fulfills the aims and scope of Metals. Presened investigations are interesting and worth to be considered for publishing. However, paper needs some improvements. My comments are listed below.

General remarks:

  • Please add the quantitative results into the abstract.
  • None of presented references have been published after 2020. You should add some positions published in 2021 and 2022. Even in MDPI there are relevant papers in the field of tool shape of bobbin FSW: https://scholar.google.com/scholar?as_ylo=2021&q=mdpi+tool+shape+friction+stir+welding&hl=pl&as_sdt=0,5
  • Moreover, I can suggest some works in the field of Bobbin FSW by Dr. Tamadon: https://www.researchgate.net/profile/Abbas-Tamadon

Introduction:

  • Fig. 1 - if this picture was takken from other paper, please add reference.
  • I propose to add some information about used Al alloy. Which methods were used to join this material, which problems were obserwed. It allow to describe the necessity of BTFSW method of joinint.
  • The novelty should be more described. It is not clear what new has been proposed in your work. Please show potential advantages from your idea.

Materials and Methods:

  • You should describe, why these parameters were used. Have you performed preeliminary investigations or they were taken from literature.
  • Fig. 4 - why this kind of plan was used? The qualification of welding technology standards suggests different order (ofcourse they include full length of the joing).

Results and discussion:

  • Results were described well. However, my biggest concern is about discussion. I cannot find any scientific discussion with comparision results with other scientific papers. You should compare your findings with other scientists. It allows to underline the biggest advantages from your work. Moreover, it will prove the novelty of your investigations. You could mark the advantages of BT with angle to different shapes.

Conclusions:

  • Please support conclusions with the quantitative results.

Author Response

General remarks:

Response to reviewers

Reviewer 2

First of all, we would like to thank you for finding the time to review our manuscript. We appreciate all comments and suggestions, and we hope we were able to address them all properly.

  • Please add the quantitative results into the abstract.

Thank you for your remark. We added quantitative results in the abstract.

  • None of presented references have been published after 2020. You should add some positions published in 2021 and 2022. Even in MDPI there are relevant papers in the field of tool shape of bobbin FSW: https://scholar.google.com/scholar?as_ylo=2021&q=mdpi+tool+shape+friction+stir+welding&hl=pl&as_sdt=0,5

Moreover, I can suggest some works in the field of Bobbin FSW by Dr. Tamadon: https://www.researchgate.net/profile/Abbas-Tamadon

Thank you very much for the suggestion. We have introduced a few references for more recent dates.

Introduction:

  • 1 - if this picture was takken from other paper, please add reference.

It was adapted from two papers, so we have introduced those papers.

  • I propose to add some information about used Al alloy. Which methods were used to join this material, which problems were obserwed. It allow to describe the necessity of BTFSW method of joinint.

Thank you for your suggestion. We have added more details about Al-Mg alloy and problems during conventional FSW of it.

  • The novelty should be more described. It is not clear what new has been proposed in your work. Please show potential advantages from your idea.

The aim was to find the influence of shoulder angle, which is added at the end of the introduction.

Materials and Methods:

  • You should describe, why these parameters were used. Have you performed preeliminary investigations or they were taken from literature.

We added this explanation.

  • 4 - why this kind of plan was used? The qualification of welding technology standards suggests different order (ofcourse they include full length of the joing).

Indeed. We opted for this specimen cutting plan to place the metallographic (macro - and micro tests) closer to a less heated part of the joint, which is less convenient compared to the standard scheme. Also, we were interested in obtaining average (beginning and ending) tensile and impact properties. Bend specimens were taken from in the central part since we expected the least problems with them since the bobbin tool forms both sides of the weld. If you believe this explanation should be added, no problem.

Results and discussion:

  • Results were described well. However, my biggest concern is about discussion. I cannot find any scientific discussion with comparision results with other scientific papers. You should compare your findings with other scientists. It allows to underline the biggest advantages from your work. Moreover, it will prove the novelty of your investigations. You could mark the advantages of BT with angle to different shapes.

We gave a discussion related to tensile and bend tests, as the second is the most significant when bobbin tools are used.

Conclusions:

  • Please support conclusions with the quantitative results.

Thank you. We added some results.

Round 2

Reviewer 2 Report

Dear Authors,

Thank you for your response. I agree with your answers. The paper has been improved a lot. IT could be published in this state.

Best regards,

Reviewer